# Differences in risk factors associated with single and multiple concurrent forms of undernutrition (stunting, wasting or underweight) among children under 5 in Bangladesh: a nationally representative cross-sectional study

Mohammad Rocky Khan Chowdhury,[1] Hafiz T A Khan [ID],[2] Mamunur Rashid,[3] Russell Kabir,[4] Sazin Islam,[1] Md Shariful Islam,[1] Manzur Kader [ID] [5]

For numbered affiliations see end of article.

**Correspondence to**
Dr Manzur Kader;
manzur.kader@ki.se

## ABSTRACT

**Objectives** The study aims to differentiate the risk factors of single and multiple concurrent forms of undernutrition among children under 5 in Bangladesh.

**Design** A nationally representative cross-sectional study.

**Setting** Bangladesh.

**Respondents** Children age under 5 years of age.

**Outcome measure** This study considered two dichotomous outcomes: single form (children without single form and with single form) and multiple concurrent forms (children without multiple forms and with multiple forms) of undernutrition.

**Statistical analysis** Adjusted OR (AOR) and CI of potential risk factors were calculated using logistic regression analysis.

**Results** Around 38.2% of children under 5 in Bangladesh are suffering from undernutrition. The prevalence of multiple concurrent forms and single form of child undernutrition was 19.3% and 18.9%, respectively. The key risk factors of multiple concurrent forms of undernutrition were children born with low birth weight (AOR 3.76, 95% CI 2.78 to 5.10); children in the age group 24–35 months (AOR 2.70, 95% CI 2.20 to 3.30) and in the lowest socioeconomic quintile (AOR 2.57, 95% CI 2.05 to 3.23). In contrast, those children in the age group 24–35 months (AOR 1.94, 95% CI 1.61 to 2.34), in the lowest socioeconomic quintile (AOR 1.79, 95% CI 1.45 to 2.21) and born with low birth weight (AOR 1.52, 95% CI 1.11 to 2.08) were significantly associated with a single form of undernutrition. Parental education, father's occupation, children's age and birth order were the differentiating risk factors for multiple concurrent forms and single form of undernutrition.

**Conclusion** One-fifth of children under 5 years of age are suffering multiple concurrent forms of undernutrition, which is similar to the numbers suffering the single form. Parental education, father's occupation, children's age and birth order disproportionately affect the multiple concurrent forms and single form of undernutrition, which should be considered to formulate an evidence-based strategy for reducing undernutrition among these children.

### Strengths and limitations of this study

► To the best of our knowledge, this is the first study to differentiate the factors associated with single and multiple concurrent forms of undernutrition among children aged under 5 in Bangladesh and is based on a nationally representative, probability-based cross-sectional sample survey.

► The multistage sampling technique of the national survey (Bangladesh Demographic Health Survey 2017–2018) used in this study, including its sampling weight, helped in reducing potential selection bias.

► The study considered a wide range of variables to assess the risk factors of the single and multiple concurrent forms of undernutrition by adopting a robust analytical method.

► All three indicators, such as stunting, wasting and underweight, were used to formulate the study outcomes without considering distinct associated factors that might affect the results.

► The cross-sectional nature of the study sample does not allow for establishing causation.

## INTRODUCTION

Child undernutrition refers to deficiencies or imbalances in a child's intake of energy and/ or nutrients, infectious disease or a combination of both.[1] In 2020, approximately 149 million children worldwide aged under 5 were estimated to be stunted, with 45 million estimated to be wasted and 85 million underweight. About 45% of deaths in children are linked to these conditions.[1,2] The burden of multiple concurrent forms of undernutrition among under 5-year-old children is present in 124 countries out of 141 (88%) with 41 countries (around 29%) struggling with high levels of all three concurrent forms of

undernutrition (stunting, wasting and underweight).[3] The multiple concurrent forms of undernutrition lead to a 12-fold elevated risk of child mortality compared with healthy children.[4] Also globally, around 3% of children under 5 years were estimated to exhibit both stunting and wasting.[5] The effect of stunting, wasting and underweight on mortality through reduced muscle mass suggests that young infants and children are especially vulnerable to undernutrition, particularly the multiple concurrent forms.[6] A reduced muscle or fat mass (a common mechanism involving limbs being both thinner and shorter) increases the risk of death considerably when wasting and stunting are present in the same child.[6] Fat stimulates the necessary energy to maintain the immune system so a undernourished child with low muscle mass and reduced function of key organs, such as the heart, kidney and immune system, is more likely to die during an acute food shortage.[6] Overall, undernutrition reduces the immunological capacity of a child to defend against diseases and recurrent infections, such as diarrhoea, pneumonia, malaria, measles and AIDS, that leads to inconsistent growth and development both mentally and physically.[7 8]

Child undernutrition is a global health problem and is particularly prevalent in low-and middle-income countries. In spite of a decline over the years, the rate of undernutrition in Bangladesh is still among the highest in the world, with estimates putting around 40% of children aged under 5 in that category.[9 10] The risk factors leading to child undernutrition are still being debated in Bangladesh and other developing countries. Key risk factors are poor maternal education, low socioeconomic status, low birth weight, poor feeding practices, frequent infections, inadequate access to healthcare and water and sanitation.[7 11–13] However, these risk factors were identified using conventional disaggregated indicators (eg, stunting, wasting and underweight). The findings from these studies are not sufficient to enable formulation of complete policy initiatives as many children in Bangladesh (around 30%) and those in other low-income and middle-income countries are suffering from multiple concurrent forms of undernutrition that lead to a high mortality rate among children.[14]

Those factors associated with the single form and multiple concurrent forms of undernutrition have not been previously identified in Bangladesh. Chowdhury *et al*, tried to link several sociodemographic factors with single and multiple forms of severe undernutrition.[14] However, the study appended multiple datasets of previous national surveys that did not represent the current scenario nor the overall burden of single and multiple concurrent forms of undernutrition. Drivers or factors associated with both these forms of undernutrition have not yet been fully identified in Bangladesh or in other low-income and middle-income countries. According to recent research findings, age, sex and food insecurity are linked to multiple concurrent forms of undernutrition.[4 15 16] On the other hand, factors associated with the single form have not been widely investigated in

developing countries, including Bangladesh. Studies on single and multiple concurrent forms of undernutrition might present different results from those grounded in conventional disaggregated indicators. This study aims to differentiate between the risk factors of both the multiple concurrent forms and a single form of undernutrition among under 5-year-old children in Bangladesh using a more recent nationally representative population-based survey. This study might, therefore, be useful for policy-makers when addressing the needs of more nutritionally vulnerable children.

## METHODOLOGY
### Study setting
Bangladesh is a densely populated country of 165 million people, with 14 million of them under 5 years of age.[17] The country is divided into eight administrative divisions, namely, Barishal, Chattogram, Dhaka, Khulna, Mymensingh, Rajshahi, Rangpur and Sylhet. Each division is divided into zilas (districts) and each zila into upazilas (subdistricts). The urban areas in an upazila are further divided into wards, and these wards are further subdivided into mohallas (suburbs). Urban areas are also classified into two groups: city corporations and areas other than city corporations. Rural areas in an upazila are divided into union parishads (villages) and then further subdivided into mouzas (small villages). Overall, the divisions separate the country into rural and urban areas with 54% of households in rural areas.[10]

### Management of undernutrition in Bangladesh
The rate of undernutrition in Bangladesh is among the highest in the world. The country has developed facility-based care (therapeutic care for malnourished children with complications) and community-based care (undernourished children without complications and children who have been discharged from facility-based inpatient care) for the management of undernourished children.[18 19] In line with this, Bangladesh's government has also introduced several policies and strategies to help deal with the rate of undernutrition. For example, in 1997, the Bangladesh National Nutrition Council developed a policy called the Bangladesh National Plan of Action for Nutrition (NPAN) to improve the nutritional status and quality of life of the people of Bangladesh.[20] The NPAN was approved by the Food Planning and Monitoring Committee in August 2008 to ensure an adequate and stable supply of safe and nutritious food, to increase people's purchasing power and access to food and thereby provide proper nutrition for all individuals, especially women and children.[21] The government of Bangladesh has again taken the initiative to incorporate nutrition into public health and family planning services to improve the country's nutrition situation. The adoption of the National Nutritional Policy 2015 reflects the commitment of the State to improve the nutritional status and quality of life of the population especially for

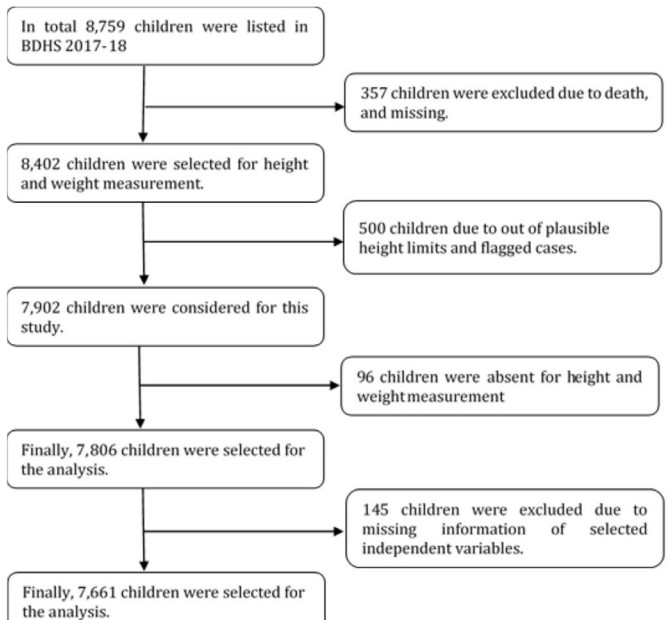

In total 8,759 children were listed in BDHS 2017-18

357 children were excluded due to death, and missing.

8,402 children were selected for height and weight measurement.

500 children due to out of plausible height limits and flagged cases.

7,902 children were considered for this study.

96 children were absent for height and weight measurement

Finally, 7,806 children were selected for the analysis.

145 children were excluded due to missing information of selected independent variables.

Finally, 7,661 children were selected for the analysis.

**Figure 1** Sample size selection. BDHS, Bangladesh Demographic Health Survey.

disadvantaged groups, including mothers, adolescent girls and children.[22] A new holistic policy called 'Food and Nutrition Security Policy' has also been adopted to promote the use of a 'nutrition lens'. This policy will help develop multisectoral interlinked interventions that will improve nutritional outcomes and cover the target period 2020–2030 in synchronisation with the Sustainable Development Goal (SDG).[23]

### Data source
After excluding missing information regarding selected variables, a total of 7661 children aged under 5, who were born in January 2013 or later, were included in this study with data taken from the Bangladesh Demographic Health Surveys (BDHS) 2017–2018 (figure 1). The BDHS 2017–2018 data are nationally representative with a 99% response rate that includes data from adults and children (both male and female) regarding demographic and social characteristics and health and nutritional indicators to monitor a wide range of populations. The data collection was started on 24 October 2017 and ended on 15 March 2018.[10] This survey was funded by the US Agency for International Development and conducted by the National Institute of Population Research and Training under the Ministry of Health and Family Welfare, Bangladesh. All survey-related issues were implemented by a Bangladeshi research organisation 'Mitra and Associate' with technical support from the Inner City Fund (ICF) International of Calverton, Maryland, USA.[10]

These surveys were based on multistage stratified sampling techniques of households. At the first stage, 675 primary sampling units (PSUs) were selected (250 PSU from urban and 425 PSU from rural) based on enumeration areas (clusters) from the census survey 2011 designed by the Bangladesh Bureau of Statistics using probability proportional to size technique. Then, a systematic sample of 30 households from each PSU was selected at the final stage using an equal probability systematic sampling technique.[10] This multistage sampling technique, including its sampling weight, helps reduce potential sampling bias.[10] In the BDHS data, sample weights were calculated in each sampling stage, each cluster and stratum were considered that had been adjusted for non-response to obtain the final standard weights.[10] In addition, all ever-married women aged 15–49 years from the preselected households were interviewed without replacement and change in the implementing stage to prevent selection bias.[10] Informed consent was obtained verbally from each participant to collect information about them and their children before enrolling in the study. A significant number of the study sample was illiterate, so verbal consent was considered the most suitable option to confirm participation.[10] Each BDHS used a standard questionnaire and the details, including sample design, data collection procedure and other issues, are discussed elsewhere.[10]

### Outcome variables and operational definitions
The outcomes of the study involved the identification of multiple concurrent forms and single form of undernutrition among children aged under 5 years. A child was considered to be undernourished or stunted (short stature for age), wasted (dangerously thin), and underweight (low weight for age) if the height-for-age, weight-for-height and weight-for-age indices were 2 SDs or more below the respective median of the WHO reference population.[24] A child was considered to have multiple concurrent forms of undernutrition when stunting and underweight and wasting and underweight, or three forms of undernutrition (stunting, wasting and underweight) were present in the same child (figure 2). Also, a child with a single form of undernutrition included a child living with stunting, wasting, or underweight only (figure 2). A detailed variable scale of measurement with missing information is clearly defined in online supplemental table 1). Children were categorised 0 for 'normal' and 1 for 'undernutrition' for each of the three indicators. After that, the values of all three indicators were added, which resulted in a score ranging from 0 to 3. The scores were again recategorised as 0 for normal, 1 for a single form of undernutrition and 2–3 for multiple concurrent forms of undernutrition. Detailed classifications and coding of outcome variables used in this study are clearly defined in online supplemental table 2).

### Independent variables
Since 1990, the UNICEF conceptual framework on undernutrition has guided interventions from multisectoral and multidimensional perspectives.[25] This framework included the basic, underlying and immediate causes of undernutrition. The basic causes address social, economic, environmental and political issues that lead to the lack of or unequal distribution of financial, human, physical, social and natural resources. The underlying

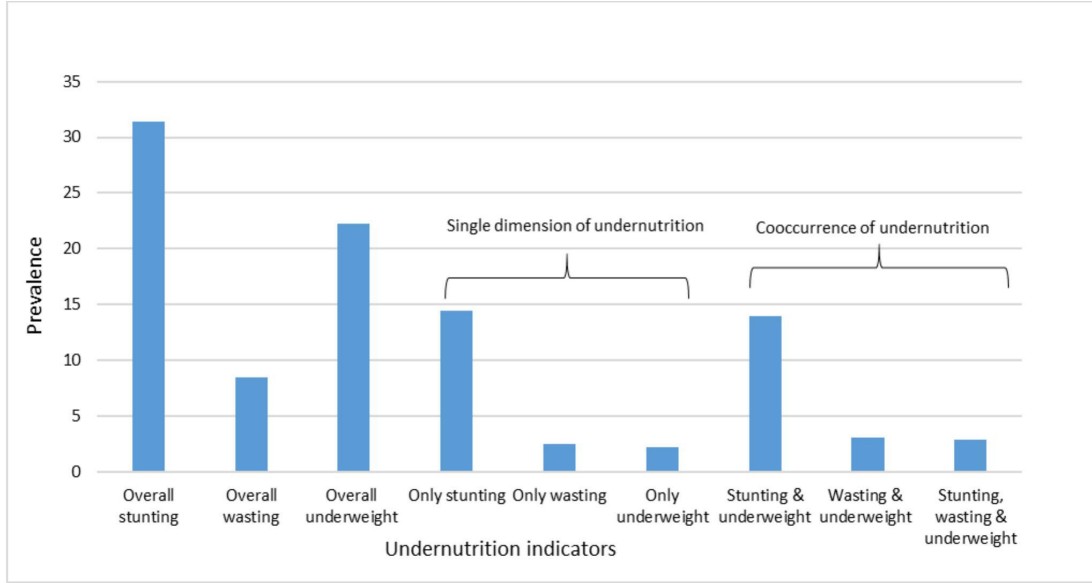

**Figure 2** Trends of the prevalence of stunting, wasting and underweight.

causes focus on household food security, adequate care and feeding practices, access to health services and residing in a healthy environment. The basic and underlying causes lead to immediate causes at the individual level through inadequate food intake and disease.[25]

The candidate variables (potential causes) considered for analysis in this study were based on relevant previous literatures.[7 26–30] These variables include maternal age in years (15–19, 20–24, 25–29, 30–34, 35–39, ≥40); parents' education (both parents uneducated, only father was uneducated when mother was educated, only mother was uneducated when father was educated, both parents educated); mother's current working status (currently not working, currently working); mother's underweight status ((normal/average (≥18.5 kg/m$^2$), underweight (<18.5 kg/m$^2$)); status of mother's antenatal and postnatal care (not received, received); mother's experience of inmate partner violence (IPV) (not experienced, experienced): a wife being beaten by partner if she went out without telling him or/and neglected the children or/and she ever argued with her partner or/and burned food or/and refused to have sex);[10] mothers' decision-making autonomy (not experienced, experienced): a woman who usually decides by herself/jointly with husband at least on her healthcare or on large household purchases or visits to family or relatives);[10] father's occupation (currently not working, manual labourer, professional and businessman); source of drinking water (improved, unimproved),[31] use of solid waste in cooking (solid, non-solid); type of toilet facility (improved, unimproved),[31] mass media exposure (no, yes: exposed to either radio, television, newspapers or magazines at least once a week); wealth index (integrating household asset ownership and access to drinking water and sanitation)[10] and place of residence (urban, rural). Moreover, various factors related to children, such as age, birth order and birthweight status,[32–34] recent morbidity status (the child

had at least one morbid condition out of diarrhoea, fever and cough in the 2 weeks preceding the survey)[10] were included. Detailed variable scales of measurement with missing information used in this study is clearly defined in online supplemental tables 3–5) .

### Patient and public involvement

Patients or the public were not involved in the design, conduct or interpretation of the study.

### Statistical analysis

The background characteristics of the children and their parents were calculated using descriptive statistics. Cross-tab analysis and bivariate analysis ($\chi^2$ test) were used to determine the prevalence/percentages and associations (p values) between the selected independent variables and outcomes. Further, all independent variables found to be significant in bivariate analysis were simultaneously entered into the stepwise (backword selection) multivariate regression models for adjustment to help identify risk factors for multiple concurrent forms and single form of undernutrition and when the base outcome (reference category) was normal (various forms of undernutrition vs normal/healthy children). The findings from logistic regression analysis were assessed using OR and CI. The significance level was set at p<0.25 (two tailed) in bivariate analyses instead of the traditional cut-off point 0.05 that can fail in identifying variables known to be important.[35] On the other hand, the significance level was set at p<0.05 (two tailed) in multivariate analyses. The data were analysed by controlling cluster (PSU) and stratum (urban area: city corporation area and other than city corporation area and rural area) with sampling weights that represent the whole country, both urban and rural areas, to ensure the precision of the estimates. Multicollinearity was checked by examining the SEs of regression coefficients in the logistic regression analyses. An

SE >2.0 indicates multicollinearity among the independent variables.[36] The SEs for the independent variables in the adjusted models for each outcome were <1, indicating an absence of multicollinearity. Stata V.14.2 (StataCorp) was used for all analyses considering the complex nature of the sampling weight of the BDHS. To adjust sampling weight in the analysis, the Stata command 'svyset' was used.

## RESULTS

Table 1 shows that approximately 7% (both parents uneducated: 3.8% and only mothers were uneducated: 3.2%) of mothers had no formal education; 15% of mothers were reported underweight; more than 90% of mothers received antenatal care; one-third (33%) did not receive postnatal care and 18% experienced IPV. Around 42% of children were from poor households with 66% living in rural areas. A similar percentage distribution of children (approximately 20%) was observed among the various age categories, and 52% of them were male (table 1).

### Prevalence of multiple concurrent forms and single form of undernutrition

In Bangladesh, the overall prevalence of child undernutrition was 38%. The overall prevalence of stunting was 31%; wasting was 8% and underweight was 22% (figure 2). The prevalence of multiple concurrent forms (from figure 1, stunting and underweight: 13.5%; wasting and underweight: 3.1%; and stunting, wasting, underweight: 2.7%) and single form (from figure 1, only stunting: 14.3%, only wasting: 2.1% and only underweight: 2.4%) of undernutrition among under 5-year-old children were approximately estimated at 19%. Further, the prevalence of multiple concurrent forms of undernutrition was significantly higher among children of parents with no formal education (37%) when compared with educated parents (17%), children of underweight mothers (about 30%), children with fathers who were manual labourers (28%), children were ≥4 birth order (about 28%) and children born with low birth weight (27%) (table 2).

The single form of undernutrition was highly prevalent among children of fathers with no formal education when mothers are educated (23%); children in the age group 12–35 months (23%); those from the poorest section of society (22%) when compared with the richest (14%); of underweight mothers and households that had no mass media exposure (21%); of fathers who were manual labourers and households with unimproved toilet facilities (21%) and children born with low birth weight (20%) (table 3).

### Risk factors

The key risk factors of multiple concurrent forms of undernutrition were children born with low birth weight (when compared with healthy weight children) (adjusted OR, AOR 3.76, 95% CI 2.78 to 5.10, p<0.001); children in the age group 24–35 months (when compared with children

| Table 1 | Background characteristics of the under 5 children | |
|---|---|---|
| **Factors** | **No** | **%** |
| Total | 7661 | 100.0 |
| Mothers' age (in years) | | |
| 15–19 | 938 | 12.2 |
| 20–24 | 2679 | 35.0 |
| 25–29 | 2146 | 28.0 |
| 30–34 | 1293 | 16.9 |
| 35–39 | 481 | 6.3 |
| ≥40 | 124 | 1.6 |
| Parents' education | | |
| Both parents were uneducated | 294 | 3.8 |
| Only father was uneducated | 865 | 11.3 |
| Only mother was uneducated | 243 | 3.2 |
| Both parents were educated | 6259 | 81.7 |
| Mother currently working | | |
| No | 4560 | 59.5 |
| Yes | 3101 | 40.5 |
| Underweight mother* | | |
| No | 6535 | 85.3 |
| Yes (<18.5 kg/m$^2$) | 1126 | 14.7 |
| Mothers received antenatal care (n=4540) | | |
| No | 363 | 8.0 |
| Yes | 4177 | 92.0 |
| Mothers received postnatal care (n=4535) | | |
| No | 1509 | 33.3 |
| Yes | 3026 | 66.7 |
| Mothers experience IPV† | | |
| No | 6264 | 81.8 |
| Yes | 1397 | 18.2 |
| Mothers' decision-making autonomy‡ | | |
| Not practised | 1083 | 14.1 |
| Practised | 6578 | 85.9 |
| Father's occupation | | |
| Currently not working | 73 | 1.0 |
| Manual labourer | 5458 | 71.2 |
| Professional | 472 | 6.2 |
| Businessman | 1658 | 21.6 |
| Source of water | | |
| Improved | 6658 | 86.9 |
| Unimproved | 1003 | 13.1 |
| Type of toilet facility | | |
| Improved | 4359 | 56.9 |
| Unimproved | 3302 | 43.1 |
| Solid waste use in cooking | | |
| No | 2218 | 28.9 |
| Yes | 5443 | 71.1 |
| Mass media exposure§ | | |
| No | 4890 | 63.8 |

Continued

**Table 1** Continued

| Factors | No | % |
|---|---|---|
| Yes | 2771 | 36.2 |
| **Wealth index¶** | | |
| Poorest | 1708 | 22.2 |
| Poorer | 1545 | 20.2 |
| Middle | 1381 | 18.0 |
| Richer | 1533 | 20.0 |
| Richest | 1494 | 19.5 |
| **Place of residence** | | |
| Urban | 2,605 | 34.0 |
| Rural | 5056 | 66.0 |
| **Children's age (in months)** | | |
| 0–11 months | 1673 | 21.8 |
| 12–23 months | 1583 | 20.7 |
| 24–35 months | 1475 | 19.3 |
| 36–47 months | 1417 | 18.5 |
| 48–59 months | 1513 | 19.7 |
| **Sex of child** | | |
| Male | 3995 | 52.2 |
| Female | 3666 | 47.8 |
| **Birth order** | | |
| One | 2902 | 37.9 |
| Two | 2507 | 32.7 |
| Three | 1297 | 16.9 |
| Four and above | 955 | 12.5 |
| **Low birth weight\*\*(n=4735)** | | |
| No | 1815 | 38.3 |
| Yes (<2.5 kg) | 326 | 6.9 |
| Not weighted | 2594 | 54.8 |
| **Currently had disease††** | | |
| No | 4043 | 52.8 |
| Yes | 3618 | 47.2 |

\*Underweight measured as <18.5 kg/m$^2$.
†A wife being beaten if she went out without telling her partner/ neglected the children/argued with her partner/burnt food/ was forced to have sex regardless of her consent.
‡A woman usually can decide by herself or jointly on her healthcare/ large household purchases/visits to family or relatives.
§Exposed television/radio/newspaper/magazine to some extent.
¶Integrating household asset ownership and access to drinking water and sanitation.
\*\*Child's weight at birth measured as ≤2.5 kg.
††Child had at least one morbid condition out of diarrhoea, fever and cough in the 2 weeks preceding the survey.
IPV, inmate partner violence.

in the age group 0–11 months) (AOR 2.70, 95% CI 2.20 to 3.30, p<0.001); children in the lowest socioeconomic quintile (when compared with the highest socioeconomic quintile) (AOR 2.57, 95% CI 2.05 to 3.23, p<0.001); parents with no formal education (when compared with educated parents) (AOR 2.03, 95% CI 1.53 to 2.71, p<0.001) and children with fathers who were currently

unemployed (when compared with businessman fathers) (AOR 1.98, 95% CI 1.07 to 3.16, p=0.021) (table 4).

The risk factors for a single form of undernutrition include children in the age group 24–35 months (when compared with children in the age group 0–11 months) (AOR 1.94, 95% CI 1.61 to 2.34, p<0.001); children in the lowest socioeconomic quintile (when compared with the highest socioeconomic quintile) (AOR 1.79, 95% CI 1.45 to 2.21, p<0.001); children born with low birth weight (when compared with healthy weight children) (AOR 1.52, 95% CI 1.11 to 2.08, p=0.008); children of fathers who are manual labourers (when compared with businessman fathers) (AOR 1.36, 95% CI 1.16 to 1.59, p<0.001); and children of fathers with no formal education when mothers are educated (when compared with educated parents) (AOR 1.35, 95% CI 1.12 to 1.63, p=0.002) (table 4).

## DISCUSSION

The prevalence of multiple concurrent forms and a single form of undernutrition among children aged under 5 in Bangladesh was nearly equal, each accounting for 19%. One out of five children suffered from both cases of undernutrition. This high figure is a concern when comparing it with developing countries from Africa and South America, such as Ethiopia, Malawi and Argentina where the prevalence of concurrent forms of undernutrition among children age under 5 stood respectively at 26%, 12% and 2%. The single form of undernutrition was respectively 23%, 39% and 5%.[37–39] Asian countries such as Yemen and India are struggling with a high burden of both forms of undernutrition (multiple concurrent forms of undernutrition: 48% and 39%, respectively; single of undernutrition: 21% and 24%, respectively).[40 41] The findings indicate that Bangladesh has not still achieved sustainable improvement in reducing child undernutrition. One reason could be poor understanding of the multisectoral approach involving a lack of coordination among key government sectors, such as between health, agriculture, education, urban and local development to address the issue of undernutrition, and poor linkage between key institutions, for example, government institutions, academic, research and training institutions and national/international non-governmental organisations.[42]

This study reveals that mothers with no formal education in Bangladesh live in less protective environments. Three out of five children (65.4%) of mothers with no formal education and two out of five children (37.4%) experienced multiple concurrent forms of undernutrition. Also, 3 out of 10 children (27.3%) born with low birthweight experience multiple concurrent forms of undernutrition. Further, the odds of multiple concurrent forms of undernutrition were 3.37 times higher among children born with a low birth weight than the children with a healthy birth weight. Previous studies in Bangladesh, Pakistan, Nepal, Malawi, Mexico and Iran also

**Table 2** Prevalence of multiple concurrent forms of undernutrition among under 5 children

| Factors | Multiple concurrent forms of undernutrition | | | |
| | **No** | | **Yes** | |
| | No (%) | Proportion (95% CI) | No (%) | Proportion/prevalence (95% CI) |
|---|---|---|---|---|
| Total | 6133 | 80.7 (79.5,81.8) | 1528 | 19.3 (18.2 to 20.5) |
| Mothers' age (in years) | | | | |
| 15–19 | 755 | 81.1 (78.1 to 83.8) | 183 | 18.9 (16.2 to 21.9) |
| 20–24 | 2187 | 81.7 (79.9 to 83.4) | 492 | 18.3 (16.6 to 20.1) |
| 25–29 | 1697 | 79.8 (77.9 to 81.6) | 449 | 20.2 (18.4 to 22.1) |
| 30–34 | 1029 | 80.6 (77.8 to 83.2) | 264 | 19.4 (16.8 to 22.2) |
| 35–39 | 378 | 80.1 (75.8 to 83.8) | 103 | 19.9 (16.2 to 24.2) |
| ≥40 | 87 | 71.0 (61.1 to 79.2) | 37 | 29.0 (20.8 to 38.9) |
| p values ($\chi^2$ test) | 0.160 | | | |
| Parents' education | | | | |
| Both parents were uneducated | 186 | 62.6 (55.7 to 69.0) | 108 | 37.4 (31.0 to 44.3) |
| Only father was uneducated | 644 | 74.4 (70.9 to 77.7) | 221 | 25.6 (22.3 to 29.1) |
| Only mother was uneducated | 168 | 72.0 (65.0 to 78.1) | 75 | 28.0 (21.9 to 35.0) |
| Both parents were educated | 5135 | 82.7 (81.5 to 83.8) | 1124 | 17.3 (16.2 to 18.5) |
| p values ($\chi^2$ test) | <0.001 | | | |
| Mother currently working | | | | |
| No | 3715 | 82.2 (80.7 to 83.6) | 845 | 17.8 (16.4 to 19.3) |
| Yes | 2418 | 78.4 (76.6 to 80.1) | 683 | 21.6 (19.9 to 23.4) |
| p values ($\chi^2$ test) | 0.0004 | | | |
| Underweight mother | | | | |
| No | 5353 | 82.3 (81.0 to 83.5) | 1182 | 17.7 (16.5 to 19.0) |
| Yes (<18.5 kg/m$^2$) | 780 | 70.5 (67.5 to 73.3) | 346 | 29.5 (26.7 to 32.5) |
| p values ($\chi^2$ test) | <0.001 | | | |
| Mothers received antenatal care | | | | |
| No | 266 | 73.8 (68.7,78.4) | 97 | 26.2 (21.6 to 31.3) |
| Yes | 3476 | 84.0 (82.6,85.2) | 701 | 16.0 (14.8 to 17.4) |
| p values ($\chi^2$ test) | <0.001 | | | |
| Mothers received postnatal care | | | | |
| No | 1290 | 85.8 (83.7 to 87.7) | 219 | 14.2 (12.3 to 16.3) |
| Yes | 2448 | 81.7 (80.1 to 83.3) | 578 | 18.3 (16.7 to 19.9) |
| | p=0.001 | | | |
| Mothers experienced IPV | | | | |
| No | 5055 | 81.3 (80.1 to 82.5) | 1209 | 18.7 (17.5 to 19.9) |
| Yes | 1078 | 77.8 (75.2 to 80.2) | 319 | 22.2 (19.8 to 24.8) |
| p values ($\chi^2$ test) | 0.005 | | | |
| Mothers' decision-making autonomy | | | | |
| Not practised | 852 | 79.5 (76.7 to 82.1) | 231 | 20.5 (17.9 to 23.3) |
| Practised | 5281 | 80.8 (79.5 to 82.1) | 1297 | 19.2 (17.9 to 20.5) |
| P values ($\chi^2$ test) | 0.370 | | | |
| Father's occupation | | | | |
| Currently not working | 52 | 71.9 (58.6 to 82.2) | 21 | 28.1 (17.8 to 41.4) |
| Manual labourer | 4277 | 79.1 (77.7 to 80.4) | 1181 | 20.9 (19.6 to 22.3) |
| Professional | 425 | 90.4 (87.1 to 92.9) | 47 | 9.6 (7.1 to 12.9) |
| Businessman | 1379 | 84.0 (81.8 to 86.0) | 279 | 16.0 (14.0 to 18.2) |

Continued

**Table 2** Continued

| Factors | Multiple concurrent forms of undernutrition | | | |
| --- | --- | --- | --- | --- |
| | No | | Yes | |
| | No (%) | Proportion (95% CI) | No (%) | Proportion/prevalence (95% CI) |
| p values ($\chi^2$ test) | p<0.001 | | | |
| Source of water | | | | |
| Improved | 5309 | 80.3 (79.0 to 81.6) | 1349 | 19.7 (18.4 to 21.0) |
| Unimproved | 824 | 82.9 (80.1 to 85.4) | 179 | 17.1 (14.6 to 19.9) |
| P values ($\chi^2$ test) | 0.092 | | | |
| Type of toilet facility | | | | |
| Improved | 3587 | 82.6 (81.1 to 84.0) | 772 | 17.4 (16.0 to 18.9) |
| Unimproved | 2546 | 78.1 (76.2 to 79.9) | 756 | 21.9 (20.1 to 23.8) |
| | 0.0001 | | | |
| Solid waste use in cooking | | | | |
| No | 1888 | 84.7 (82.7 to 86.5) | 330 | 15.3 (13.5 to 17.3) |
| Yes | 4245 | 78.9 (77.5 to 80.3) | 1198 | 21.1 (19.7 to 22.5) |
| p values ($\chi^2$ test) | <0.001 | | | |
| Mass media exposure | | | | |
| No | 2095 | 76.2 (74.2 to 78.1) | 676 | 23.8 (21.9 to 25.8) |
| Yes | 4038 | 83.1 (81.7 to 84.3) | 852 | 16.9 (15.7 to 18.3) |
| p values ($\chi^2$ test) | <0.001 | | | |
| Wealth index | | | | |
| Poorest | 1248 | 73.6 (71.0 to 76.0) | 460 | 26.4 (24.0 to 29.0) |
| Poorer | 1172 | 77.2 (74.8 to 79.5) | 373 | 22.8 (20.5 to 25.2) |
| Middle | 1105 | 81.3 (78.6 to 83.7) | 276 | 18.7 (16.3 to 21.4) |
| Richer | 1266 | 82.8 (80.2 to 85.1) | 267 | 17.2 (14.9 to 19.8) |
| Richest | 1342 | 89.8 (87.7 to 91.5) | 152 | 10.2 (8.5 to 12.3) |
| p values ($\chi^2$ test) | p<0.001 | | | |
| Place of residence | | | | |
| Urban | 2152 | 82.8 (80.5 to 84.8) | 453 | 17.2 (15.2 to 19.5) |
| Rural | 3981 | 79.9 (78.5 to 81.3) | 1075 | 20.1 (18.7 to 21.5) |
| p values ($\chi^2$ test) | 0.032 | | | |
| Children's age (in months) | | | | |
| 0–11 months | 1454 | 88.0 (86.2 to 89.6) | 219 | 12.0 (10.4 to 13.8) |
| 12–23 months | 1293 | 82.7 (80.5 to 84.7) | 290 | 17.3 (15.3 to 19.5) |
| 24–35 months | 1141 | 77.6 (75.0 to 80.1) | 334 | 22.4 (19.9 to 25.0) |
| 36–47 months | 1097 | 77.7 (75.0 to 80.2) | 320 | 22.3 (19.8 to 25.0) |
| 48–59 months | 1148 | 76.1 (73.6 to 78.5) | 365 | 23.9 (21.5 to 26.4) |
| p values ($\chi^2$ test) | <0.001 | | | |
| Sex of child | | | | |
| Male | 3188 | 80.4 (78.8 to 82.0) | 807 | 19.6 (18.0 to 21.2) |
| Female | 2945 | 80.9 (79.3 to 82.4) | 721 | 19.1 (17.6 to 20.7) |
| p values ($\chi^2$ test) | 0.645 | | | |
| Birth order | | | | |
| One | 2385 | 82.3 (80.5 to 83.9) | 517 | 17.7 (16.1 to 19.5) |
| Two | 2043 | 82.1 (80.3 to 83.7) | 464 | 17.9 (16.3 to 19.7) |
| Three | 1023 | 80.4 (78.0 to 82.6) | 274 | 19.6 (17.4 to 22.0) |
| Four and above | 682 | 72.4 (68.8 to 75.8) | 273 | 27.6 (24.2 to 31.2) |

Continued

**Table 2**  Continued

| Factors | Multiple concurrent forms of undernutrition | | | |
| | No | | Yes | |
| | No (%) | Proportion (95% CI) | No (%) | Proportion/prevalence (95% CI) |
|---|---|---|---|---|
| p values ($\chi^2$ test) | <0.001 | | | |
| Low birth weight | | | | |
| No | 1618 | 89.5 (87.7 to 91.0) | 197 | 10.5 (9.0 to 12.3) |
| Yes (<2.5 kg) | 230 | 72.7 (67.1 to 77.7) | 96 | 27.3 (22.3 to 32.9) |
| Not weighted | 2043 | 79.7 (77.8 to 81.4) | 551 | 20.3 (18.6 to 22.2) |
| p values ($\chi^2$ test) | <0.001 | | | |
| Currently had disease | | | | |
| No | 3277 | 81.5 (79.9 to 82.9) | 766 | 18.5 (17.1 to 20.1) |
| Yes | 2856 | 79.8 (78.1 to 81.4) | 762 | 20.2 (18.6 to 21.9) |
| p values ($\chi^2$ test) | 0.103 | | | |

No undernutrition for multiple concurrent forms includes healthy children and children with single form of undernutrition.
IPV, inmate partner violence.

reported children born with low birth weight were more likely to experience malnutrition regardless of multiple concurrent forms and single form.[43–47] Children born with low birth weight generally increase their height and weight by small increments and thus they may remain shorter and lighter and might be severely malnourished without adequate nutritional support.[48] Children with low birth weights are often born to uneducated and underweight mothers.[49 50] Maternal/parental illiteracy is associated with poor maternal healthcare access, and caregiving provided to children often leads to an adverse nutritional outcome for mothers and children.[51] This study shows that the odds of multiple concurrent forms of undernutrition were, respectively, 2.03 and 1.91 times higher among children of mothers with no formal education and among underweight mothers. Numerous studies assessed these factors as key determinants of child undernutrition using disaggregated indicators and need more assessment to estimate the magnitude of risks in case of multiple concurrent forms and single form of undernutrition.[7 11]

Other key risk factors of multiple concurrent forms of undernutrition include children in the older age group (3 years and above); those in the lowest socioeconomic quintile; have fathers who are currently unemployed that differs by a single form of undernutrition, to some extent, additionally with children in the younger age group (3 years and below); have mothers with no formal education (when fathers are educated) and fathers that are labourers. An assessment of the risk difference between educated and uneducated parents and the highest and lowest socioeconomic quintile for multiple concurrent forms and a single form of undernutrition indicates higher educational and wealth inequalities for the multiple concurrent forms of undernutrition. Further, the level of parental education and economic hardship is

not markedly responsible for suffering from a single form of undernutrition among children aged under 5. Earlier studies in Bangladesh and other South Asian countries highlighted parental education and wealth index as key risk factors of child undernutrition regardless of multiple concurrent forms and single form. Still, the studies need more precise estimation that could help policy makers introduce context-specific actions for reducing undernutrition.[7 11 28 30 52] Women who have low social status and less education have very limited influence on household decision-making that might influence child nutritional status. On the other hand, a large community in Bangladesh are concerned with broader issues of chronic poverty, and this would not reflect the improvement of undernutrition at a community level.[52] To reduce under 5-year-old child undernutrition, it is recommended that the participation of women and girls in education is increased, especially in rural areas, minimising socioeconomic inequality by raising the income-generating sectors and strengthening the nutrition-specific health programmes.

Children in the older age group (3 years and above) have a high risk of multiple concurrent forms of undernutrition, while children in the younger age group (3 years and below) were more likely to have a single form of undernutrition. Many studies conducted in Bangladesh, Nepal, Pakistan, Ethiopia and Congo have consistently addressed age and older children as having a higher risk of being undernourished without classifying the two conditions.[26–30] In India, younger children were found to have a higher likelihood of being undernourished.[53] Older children receive less protection in times of adverse nutritional outcomes due to poor per capita food distribution among family members in large households when they tend to be allocated less food than will meet their energy requirements with regards to their age.[54] The risk of poor parental education on multiple concurrent forms

**Table 3** Prevalence of single form of undernutrition among under 5 children

| Factors | Single form of undernutrition | | | |
| | **No** | | **Yes** | |
| | **No (%)** | **Proportion (95% CI)** | **No (%)** | **Proportion/prevalence (95% CI)** |
|---|---|---|---|---|
| Total | 6205 | 81.1 (80.1 to 82.1) | 1456 | 18.9 (17.9 to 19.9) |
| Mothers' age (in years) | | | | |
| 15–19 | 745 | 79.9 (76.8 to 82.7) | 193 | 20.1 (17.3 to 23.2) |
| 20–24 | 2174 | 81.2 (79.4 to 82.8) | 505 | 18.8 (17.2 to 20.6) |
| 25–29 | 1759 | 82.0 (80.1 to 83.8) | 387 | 18.0 (16.2 to 19.9) |
| 30–34 | 1031 | 79.9 (77.4 to 82.1) | 262 | 20.1 (17.9 to 22.6) |
| 35–39 | 391 | 81.0 (76.7 to 84.7) | 90 | 19.0 (15.3 to 23.3) |
| ≥40 | 105 | 85.7 (77.7 to 91.2) | 19 | 14.3 (8.8 to 22.3) |
| p values ($\chi^2$ test) | 0.533 | | | |
| Parents' education | | | | |
| Both parents were uneducated | 233 | 80.9 (75.5 to 85.3) | 61 | 19.1 (14.7 to 24.5) |
| Only father was uneducated | 666 | 77.0 (73.9 to 79.8) | 199 | 23.0 (20.2 to 26.1) |
| Only mother was uneducated | 191 | 78.2 (71.3 to 83.8) | 52 | 21.8 (16.2 to 28.7) |
| Both parents were educated | 5115 | 81.8 (80.6 to 82.9) | 1144 | 18.2 (17.1 to 19.4) |
| p values ($\chi^2$ test) | 0.017 | | | |
| Mother currently working | | | | |
| No | 3704 | 81.5 (80.0 to 82.8) | 856 | 18.5 (17.2 to 20.0) |
| Yes | 2501 | 80.5 (78.9 to 82.1) | 600 | 19.5 (17.9 to 21.1) |
| p values ($\chi^2$ test) | 0.396 | | | |
| Underweight mother | | | | |
| No | 5323 | 81.5 (80.3 to 82.6) | 1212 | 18.5 (17.4 to 19.7) |
| Yes (<18.5 kg/m$^2$) | 882 | 78.8 (76.1 to 81.2) | 244 | 21.2 (18.8 to 23.9) |
| p values ($\chi^2$ test) | 0.053 | | | |
| Mothers received antenatal care | | | | |
| No | 279 | 77.5 (72.5 to 81.8) | 84 | 22.5 (18.2 to 27.5) |
| Yes | 3334 | 79.8 (78.5 to 81.2) | 843 | 20.2 (18.8 to 21.5) |
| p values ($\chi^2$ test) | 0.327 | | | |
| Mothers received postnatal care | | | | |
| No | 1210 | 80.5 (78.1 to 82.8) | 299 | 19.5 (17.2 to 21.9) |
| Yes | 2398 | 79.2 (77.6 to 80.7) | 628 | 20.8 (19.3 to 22.4) |
| | 0.335 | | | |
| Mothers experienced IPV | | | | |
| No | 5071 | 81.2 (80.0 to 82.3) | 1193 | 18.8 (17.7 to 20.0) |
| Yes | 1134 | 80.8 (78.5 to 82.9) | 263 | 19.2 (17.1 to 21.5) |
| p values ($\chi^2$ test) | 0.744 | | | |
| Mothers' decision-making autonomy | | | | |
| Not practised | 881 | 82.2 (79.6 to 84.5) | 202 | 17.8 (15.5 to 20.4) |
| Practised | 5324 | 80.9 (79.8 to 82.0) | 1254 | 19.1 (18.0 to 20.2) |
| p values ($\chi^2$ test) | 0.377 | | | |
| Father's occupation | | | | |
| Currently not working | 63 | 87.0 (76.2 to 93.3) | 10 | 13.0 (6.7 to 23.8) |
| Manual labourer | 4347 | 79.5 (78.3 to 80.6) | 1111 | 20.5 (19.4 to 21.7) |
| Professional | 414 | 89.0 (85.6 to 91.6) | 58 | 11.0 (8.4 to 14.4) |

Continued

**Table 3** Continued

| Factors | Single form of undernutrition | | | |
| | No | | Yes | |
| | No (%) | Proportion (95% CI) | No (%) | Proportion/prevalence (95% CI) |
|---|---|---|---|---|
| Businessman | 1381 | 84.3 (82.2 to 86.3) | 277 | 15.7 (13.7 to 17.8) |
| p values ($\chi^2$ test) | <0.001 | | | |
| Source of water | | | | |
| Improved | 5399 | 81.1 (80.0 to 82.2) | 1259 | 18.9 (17.8 to 20.0) |
| Unimproved | 806 | 80.9 (78.2 to 83.3) | 197 | 19.1 (16.7 to 21.8) |
| p values ($\chi^2$ test) | 0.857 | | | |
| Type of toilet facility | | | | |
| Improved | 3583 | 82.3 (80.9 to 83.6) | 776 | 17.7 (16.4 to 19.1) |
| Unimproved | 2622 | 79.5 (78.0 to 81.0) | 680 | 20.5 (19.0 to 22.0) |
| | 0.006 | | | |
| Solid waste use in cooking | | | | |
| No | 1846 | 83.2 (81.3 to 84.9) | 372 | 16.8 (15.1 to 18.7) |
| Yes | 4359 | 80.2 (79.0 to 81.4) | 1084 | 19.8 (18.6 to 21.0) |
| p values ($\chi^2$ test) | 0.008 | | | |
| Mass media exposure | | | | |
| No | 2175 | 78.8 (77.0 to 80.4) | 596 | 21.2 (19.6 to 23.0) |
| Yes | 4030 | 82.4 (81.0 to 83.6) | 860 | 17.6 (16.4,19.0) |
| p values ($\chi^2$ test) | 0.0009 | | | |
| Wealth index | | | | |
| Poorest | 1324 | 77.9 (75.7 to 79.9) | 384 | 22.1 (20.1 to 24.3) |
| Poorer | 1213 | 79.0 (76.7 to 81.2) | 332 | 21.0 (18.8 to 23.3) |
| Middle | 1136 | 82.3 (80.1 to 84.4) | 245 | 17.7 (15.6 to 19.9) |
| Richer | 1246 | 81.2 (78.9 to 83.3) | 287 | 18.8 (16.7 to 21.1) |
| Richest | 1286 | 85.8 (83.5 to 87.9) | 208 | 14.2 (12.1 to 16.5) |
| p values ($\chi^2$ test) | <0.001 | | | |
| Place of residence | | | | |
| Urban | 2173 | 84.0 (82.2 to 85.7) | 432 | 16.0 (14.3 to 17.8) |
| Rural | 4032 | 80.1 (78.8 to 81.3) | 1024 | 19.9 (18.7 to 21.2) |
| p values ($\chi^2$ test) | 0.0004 | | | |
| Children's age (in months) | | | | |
| 0–11 months | 1406 | 83.8 (81.9 to 85.6) | 267 | 16.2 (14.4 to 18.1) |
| 12–23 months | 1215 | 77.1 (74.7 to 79.4) | 368 | 22.9 (20.6 to 25.3) |
| 24–35 months | 1139 | 77.1 (74.5 to 79.6) | 336 | 22.9 (20.4 to 25.5) |
| 36–47 months | 1137 | 80.7 (78.3 to 82.9) | 280 | 19.3 (17.1 to 21.7) |
| 48–59 months | 1308 | 86.7 (84.6 to 88.5) | 205 | 13.3 (11.5 to 15.4) |
| p values ($\chi^2$ test) | 0.001 | | | |
| Sex of child | | | | |
| Male | 3241 | 81.0 (79.6 to 82.4) | 754 | 19.0 (17.6 to 20.4) |
| Female | 2964 | 81.2 (79.7 to 82.6) | 702 | 18.8 (17.4 to 20.3) |
| p values ($\chi^2$ test) | =0.863 | | | |
| Birth order | | | | |
| One | 2377 | 81.9 (80.3 to 83.4) | 525 | 18.1 (16.6 to 19.7) |

Continued

**Table 3** Continued

| Factors | Single form of undernutrition | | | |
|---|---|---|---|---|
| | No | | Yes | |
| | No (%) | Proportion (95% CI) | No (%) | Proportion/prevalence (95% CI) |
| Two | 2040 | 81.4 (79.6 to 83.1) | 467 | 18.6 (16.9 to 20.4) |
| Three | 1032 | 79.4 (76.8 to 81.8) | 265 | 20.6 (18.2 to 23.2) |
| Four and above | 756 | 80.1 (77.3 to 82.6) | 199 | 19.9 (17.4 to 22.7) |
| p values ($\chi^2$ test) | 0.285 | | | |
| Low birth weight | | | | |
| No | 1502 | 82.4 (80.2 to 84.3) | 313 | 17.6 (15.7 to 19.8) |
| Yes (<2.5 kg) | 260 | 79.7 (74.2 to 84.3) | 66 | 20.3 (15.7 to 25.8) |
| Not weighted | 2001 | 77.5 (75.8 to 79.1) | 593 | 22.5 (20.9 to 24.2) |
| p values ($\chi^2$ test) | 0.002 | | | |
| Currently had disease | | | | |
| No | 3251 | 80.9 (79.5 to 82.3) | 792 | 19.1 (17.7 to 20.5) |
| Yes | 2954 | 81.3 (79.8 to 82.7) | 664 | 18.7 (17.3 to 20.2) |
| p values ($\chi^2$ test) | 0.699 | | | |

No undernutrition for single form includes healthy children and children with multiple concurrent forms of undernutrition.
IPV, inmate partner violence.

of undernutrition differs by paternal illiteracy (when mothers are educated) on a single form of undernutrition in identifying the impact of education on undernutrition. This study shows that paternal education was associated with a single form of undernutrition among children under 5 in Bangladesh. Evidence suggests that the impact of paternal education on undernutrition has not been widely addressed while maternal education has a more significant effect on child undernutrition than paternal education.[53] Parental education is associated with positive attitudes and behaviours that can reduce the incidence of child undernutrition to some extent. It needs to be recognised that the complex nature of modern healthcare messages and behavioural changes will not be effectively understood or practised by mothers with only the most basic literacy and numeracy skills.[55] Reducing the gender gap for enrolment in universal upper secondary education may help towards tackling the burden of all undernutrition.

This study also indicates that children of unemployed fathers are more likely to experience multiple concurrent forms of undernutrition; conversely, children of fathers who were labourers are less protected in the case of a single form of undernutrition. Additionally, 3 out of 10 children experienced multiple concurrent forms of undernutrition, with 21% experiencing a single form. The effects of paternal unemployment on child undernutrition have not been widely explored, except for a study conducted in China that found it had a significant impact.[56] Conversely, those children of fathers that had low paid jobs (labourer, farmer, etc) had a greater risk of suffering undernutrition.[11 14] A loss of employment or low paid jobs are typically associated with a lower income

available for spending on market goods, including healthcare, non-household childcare and nutritious consumption.[56] Increasing investment in healthcare at the domestic level could help in tackling undernutrition.

This study strongly suggests that policy implications or interventions from the perspectives of developing countries, such as Bangladesh, are needed to successfully prevent and treat both forms of undernutrition, particularly multiple concurrent forms and based on evidence and a broad-spectrum. First, factors associated with undernutrition should be considered in order to formulate an evidence-based strategy for reducing undernutrition, with the emphasis on the multiple concurrent forms. Routine national and subnational level nutrition surveys, such as Multiple Indicator Cluster Surveys, need to be modified to include multiple concurrent forms to aid in the development of programmes and policy decision making.[57] Routine monitoring of the prevalence of multiple concurrent forms would also be required to inform effective detection and treatment.[57] Community engagement and screening for multiple concurrent forms of undernutrition could also be expanded in innovative ways by enrolling additional expertise and resources.[58] Innovative and early markers should be developed to predict, identify, and monitor children for short-term and long-term consequences.[59] Maternal factors from adolescence through pregnancy need to be scrutinised in order to identify those that can adversely affect utero and postnatal children living with three forms of undernutrition.[59] Therapeutic interventions (eg, ready-to-use therapeutic foods) must be reviewed and adjusted to ensure that children at the highest risk of mortality are included due to multiple concurrent forms of undernutrition. Comprehensive nutrition programmes must be developed for pursuing

**Table 4** Risk factors of multiple concurrent forms and single form of under 5 child undernutrition (results of stepwise logistic regression with backword selection)

| Factors | Multiple concurrent forms | | Single form | |
| --- | --- | --- | --- | --- |
| | AOR (95% CI) | P values | AOR (95% CI) | P values |
| Parents' education*†‡§ | | | | |
| Both parents were uneducated | 2.03 (1.53 to 2.71) | <0.001 | 1.33 (0.95 to 1.85) | 0.095 |
| Only father was uneducated | 1.37 (1.14 to 1.65) | 0.001 | 1.35 (1.12 to 1.63) | 0.002 |
| Only mother was uneducated | 1.57 (1.14 to 2.16) | 0.006 | 1.37 (0.98 to 1.91) | 0.067 |
| Both parents were educated | 1.00 | | 1.00 | |
| Underweight mothers*†‡§ | | | | |
| No | 1.00 | | 1.00 | |
| Yes | 1.91 (1.62 to 2.25) | <0.001 | 1.30 (1.09 to 1.55) | 0.003 |
| Father's occupation*†‡§ | | | | |
| Currently not working | 1.98 (1.11 to 3.54) | 0.021 | 1.11 (0.53 to 2.31) | 0.784 |
| Manual labourer | 1.25 (1.06 to 1.46) | 0.007 | 1.36 (1.16 to 1.59) | <0.001 |
| Professional | 0.76 (0.53 to 1.09) | 0.139 | 0.71 (0.50 to 1.01) | 0.053 |
| Businessman | 1.00 | | 1.00 | |
| Wealth index*†‡§ | | | | |
| Poorest | 2.57 (2.05 to 3.23) | <0.001 | 1.79 (1.45 to 2.21) | <0.001 |
| Poorer | 2.33 (1.86 to 2.92) | <0.001 | 1.63 (1.33 to 2.01) | <0.001 |
| Middle | 1.83 (1.46 to 2.30) | <0.001 | 1.26 (1.02 to 1.55) | 0.035 |
| Richer | 1.79 (1.43 to 2.25) | <0.001 | 1.42 (1.16 to 1.74) | 0.001 |
| Richest | 1.00 | | 1.00 | |
| Children's age (in months)*†‡§ | | | | |
| 0–11 months | 1.00 | | 1.00 | |
| 12–23 months | 1.78 (1.45 to 2.19) | <0.001 | 1.76 (1.46 to 2.10) | <0.001 |
| 24–35 months | 2.70 (2.20 to 3.30) | <0.001 | 1.94 (1.61 to 2.34) | <0.001 |
| 36–47 months | 2.51 (2.05 to 3.07) | <0.001 | 1.54 (1.27 to 1.87) | <0.001 |
| 48–59 months | 2.35 (1.93 to 2.87) | <0.001 | 0.95 (0.77 to 1.16) | 0.607 |
| Birth order*†‡§ | | | | |
| One | 1.00 | | | |
| Two | 1.02 (0.88 to 1.19) | 0.753 | | |
| Three | 1.06 (0.88 to 1.27) | 0.538 | | |
| Four and above | 1.42 (1.17 to 1.73) | 0.001 | | |
| Low birthweight*§ | | | | |
| No | 1.00 | | 1.00 | |
| Yes | 3.76 (2.78 to 5.10) | <0.001 | 1.52 (1.11 to 2.08) | 0.008 |

Baseoutocme or reference category of regression model was normal chidren (multiple concurrent forms vs normal and single form vs normal).

*Stepwise logistic regression model for multiple concurrent forms of undernutrition includes parents' education, mother's underweight status, father's occupation, use of solid waste in cooking, type of toilet facility, mass media exposure, wealth index, place of residence, age of children and birthweight status.

†Stepwise logistic regression model for single form of undernutrition includes parents' education, mother's underweight status, father's occupation, use of solid waste in cooking, type of toilet facility, mass media exposure, wealth index, place of residence and age of children.

‡Stepwise logistic regression model for multiple concurrent forms of undernutrition includes maternal age, parents' education, mother's current working status, mother's underweight status, mother's experience IPV, father's occupation, use of solid waste in cooking, type of toilet facility, mass media exposure, wealth index, place of residence, age of children and birth order.

§Stepwise logistic regression model for multiple concurrent forms of undernutrition includes maternal age, parents' education, mother's current working status mother's underweight status, status of mother's antenatal and postnatal care, mother's experience IPV, father's occupation, use of solid waste in cooking, type of toilet facility, mass media exposure, wealth index, place of residence, age of children, birth order and birthweight status.

AOR, adjusted OR; IPV, inmate partner violence.

SDG 2.2 in order to end undernutrition in all its forms by 2030.[59]

This study analysed data from a national BDHS survey 2017–2018. The findings can, therefore, be applied to the whole country and to low-income and middle-income countries whose sociodemographic characteristics and healthcare settings are similar to Bangladesh. Another strength in this study was the appropriate application of statistical tools that might help in understanding the findings. The multistage sampling technique, for instance, including its sampling weight, helped reduce potential selection bias. There were some limitations. For example, the cross-sectional nature of the data did not establish a causal relationship between risk factors and outcome variables. All three indicators (eg, stunting, wasting and underweight) were used to formulate the study outcomes (single and multiple concurrent forms of undernutrition) without considering distinct associated factors that might affect the results. This study did not control other important indicators, such as a child's diet pattern, parents' behaviour of parents or immunisation and ethnicity, due to lack of availability and missing values across the surveys. Another limitation arises due to recall bias or information bias as a result of self-reporting of age, education, occupation and household assets for example.

## CONCLUSION

One out of five children in Bangladesh aged under 5 years is suffering multiple concurrent forms and a single form of undernutrition. Children born with low birth weight, in the lowest socioeconomic quintile, and in the older age group (3 years and above) were identified as key risk factors for multiple concurrent forms of undernutrition. Those children in the younger age group (3 years and below), and mothers with no formal education had significant effects on the single dimension of undernutrition. Parental education, father's occupation, age and birth order are the main differentiating risk factors between multiple concurrent forms and single form of undernutrition. The findings highlight factors that should be considered in order to formulate an evidence-based strategy to reduce undernutrition among under 5-year-old children. In addition, concerted efforts are essential for developing strong collaboration among different sectors, such as between government, non-government, educational, social, cultural and religious institutions, in order to improve nutritional status. The findings from this study may be transferable to other low-income and middle-income countries but further study is warranted in this regard.

**Author affiliations**
[1]Department of Public Health, First Capital University of Bangladesh, Chuadanga, Bangladesh
[2]College of Nursing, Midwifery and Healthcare, University of West London, Brentford, London, UK
[3]Department of Public Health and Sports Sciences, University of Gävle, Gavle, Gävleborg, Sweden
[4]School of Allied Health, Faculty of Health, Education, Medicine, and Social Care, Anglia Ruskin University, Chelmsford, London, UK
[5]Institute of Environmental Medicine, Karolinska Institutet, Stockholm, Sweden

**Acknowledgements** The authors thank the MEASURE DHS project for their support and free access to the original data and the Department of Public Health, First Capital University of Bangladesh, enabling the research scope. The authors are also thankful to the Oxford Institute of Population Ageing, University of Oxford, UK for academic support.

**Contributors** MRKC conceptualised the basic idea for the study, performed the statistical analysis together with MSI and RK. MRKC and SI prepared data for analysis. MRKC and MK prepared the first draft of the manuscript. HTAK and MR critically revised the manuscript for intellectual content. All authors have reviewed and approved the final manuscript. MRKC is a guarantor of the study and responsible for the conduct of the study, have access to the data and control the decision to publish.

**Funding** The authors have not declared a specific grant for this research from any funding agency in the public, commercial or not-for-profit sectors.

**Competing interests** None declared.

**Patient consent for publication** Not applicable.

**Ethics approval** The BDHS was reviewed and approved by the ICF Macro Institutional Review Board (USA), which complies with all of the requirements of 45 CFR 46 'Protection of Human Subjects'. The Bangladesh DHS was also reviewed and approved by the National Research Ethics Committee of the Bangladesh Medical Research Council (Dhaka, Bangladesh). Informed consent was obtained verbally from each participant and their intimate partners (all ever-married women aged 15–49 years old) prior to being enrolled in the study. A significant number of the study sample was illiterate, so verbal consent was considered the most suitable option to confirm participation. The BDHS surveys also included samples of very young children (under 5-year-old children either born in 2014 or later) in the data collection, and so mothers of these samples were asked to provide verbal consent on behalf of their children.

**Provenance and peer review** Not commissioned; externally peer reviewed.

**Data availability statement** Data are available on reasonable request. The datasets used and/or analysed during the current study are available from the corresponding author on reasonable request.

**ORCID iDs**
Hafiz T A Khan http://orcid.org/0000-0002-1817-3730
Manzur Kader http://orcid.org/0000-0001-8181-648X

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
