## [Reviewer comments · BMJ Open]

ARTICLE DETAILS

TITLE (PROVISIONAL)	Differences in risk factors associated with single and multiple concurrent forms of undernutrition (stunting, wasting, or underweight) among children under 5 in Bangladesh: A nationally representative cross-sectional study
AUTHORS	Chowdhury, Mohammad; Khan, Hafiz; Rashid, Mamunur; Kabir, Russell; Islam, Sazin; Shariful Islam, Md.; Kader, Manzur

VERSION 1 – REVIEW

REVIEWER	Ryckman, Theresa Stanford University
REVIEW RETURNED	26-May-2021

GENERAL COMMENTS	Bmjopen-2021-052814 Review This is a well-written study comparing the factors associated with a single dimension of undernutrition against the factors associated with co-occurrence of multiple dimensions of undernutrition among children under 5 in Bangladesh. I would like to especially thank the authors for writing an extremely clear Methods section (with useful references to more details in their supplement). Major Comments (in order of importance) The stated purpose of this study is to compare factors associated with a single dimension of undernutrition against factors associated with cooccurrence. However, this is not exactly what the analytical strategy does – the current analysis actually compares factors associated with single dimension vs. no undernutrition and, separately, factors associated with cooccurrence vs. no undernutrition, not factors associated with single dimension vs. cooccurrence. It is not clear from the analysis whether the differences between single dimension and cooccurrence are statistically significant, and the presentation of the multivariate analysis (Table 3) as adjusted odds ratios also makes it more difficult to compare single vs. cooccurrence, since the table presumably shows odds compared to the reference group of no undernutrition. If the purpose of the study is to compare single vs. occurrence, a different analytical strategy that can incorporate multiple/categorical outcomes, such as multinomial regression, should be considered. Selecting only significant variables from bivariate analysis for multivariate analysis is a commonly-used technique for variable selection, but it can cause potentially important variables to be left out (e.g. variables that are only revealed to be substantive/significant when another variable is controlled for). I urge the authors to consider more robust selection techniques (for
--

example several techniques are discussed here: <https://scfbm.biomedcentral.com/articles/10.1186/1751-0473-3-17>), either to replace the main multivariate analysis (this would be preferable) or at the very least as a sensitivity analysis to include in the supplement.

At the very least, if the authors keep the current approach or move it to the appendix, $p < 0.05$ is a very harsh cutoff for variable selection, and it would be better to use a higher threshold.

The current approach also results in the multivariate analysis for single occurrence using a different set of independent variables than the multivariate analysis for cooccurrence, which seems problematic (e.g. it is difficult to conclude that mother underweight is associated only with cooccurrence and not single dimension when it is not even included in the multivariate analysis for single dimension).

Given that other studies have found that distinct factors are associated with stunting vs. wasting vs. underweight, it seems odd to lump the three together as “single dimension” in the analysis. Similarly, might there be different factors associated with being both stunting and underweight vs. being both wasted and underweight vs. all three? I would like to see more discussion of why this methodological choice was made. If there was not a strong reason for doing this, the authors might want to consider assessing each factors separately (e.g. having 6 categories of undernutrition instead of the 2 currently used).

Several variables are missing for over half of the sample (e.g. low birthweight, whether the mother received postnatal or antenatal care). By default, if those variables are included in the logistic regression, missing observations will be dropped entirely, meaning that the logistic regression analysis drops about half of the sample. If the authors used a method for alternative handling of these missing data this should be discussed in the methods. If not, the authors should consider whether/demonstrate that the reduced sample is still nationally representative and does not have substantially different characteristics compared to the full sample. If these conditions do not hold, as a sensitivity analysis the authors should run the analysis on the full sample and remove the independent variables for which there are many missing observations.

Table 3 should state the sample size to further clarify whether these observations were dropped from the analysis.

I would like to see more discussion of the importance of assessing factors associated with single dimension vs. co-occurrence of undernutrition. For example, should policy measures target co-occurrence and thus prioritize addressing those risk factors associated with co-occurrence over those associated with a single dimension of undernutrition?

Minor Comments

Abstract:

This is probably just a typo (as I think all results in the abstract are adjusted odds ratios), but it currently reads as though all of the results are presented as odds ratios except for age group 12-23 months, which appears to be an adjusted odds ratio

Methods:

First paragraph states that the BDHS “includes data from Bangladeshi adults (both male and female)” – emphasis on adults seems odd given that the focus of this study is on children and data were collected for children too (e.g. anthropometric measurements)

	Wealth index incorporates access to drinking water and sanitation, both of which are included in the analysis as distinct independent variables. It would be better to separate out these components so that the independent variables are not overlapping/redundant – for example by considering only household asset ownership and not access to drinking water/sanitation in the wealth index. Generally, clustering (e.g. of standard errors) is distinct from weighting (e.g. using survey weights). Can the authors please clarify this sentence in the statistical analysis subsection of methods? “The data were analyzed by considering cluster effects using survey weights that represent the whole country, both urban and rural areas, to ensure the precision of the estimates” Mother’s age at the child’s birth might be a more relevant variable to include in the analysis (instead of mother’s age when surveyed) Results: Table 2 should also include the reference group (no undernutrition) It would be helpful to add a brief discussion of Figure 2 to the results section. Discussion: 2nd sentence “This is a bit worrying figure when comparing it with countries...” Why is this worrying? The burden in Bangladesh is lower than many of those listed (such as India, Yemen, Ethiopia) and this seems like a very random selection of countries to compare against. Requires further explanation. “Four out of ten children of mothers with no formal education experience a cooccurrence of undernutrition.” and “Further, children born with low birth weight had a 240% higher risk of having cooccurrence of undernutrition” – where do these numbers come from? They don’t appear to be consistent with Table 2. For example, wouldn’t 654.5 (37.4% + 28.0%) of children of mothers with no formal education experience cooccurrence? Later - “This study shows that children of mothers with no formal education and underweight mothers were 1.78 and 1.51 times more likely to experience cooccurrence of undernutrition” This sentences conflates relative risks with odds ratios because it comes from the adjusted analysis, which presents odds ratios. Thus it would be correct to say that “the odds of cooccurrence are 1.78 times higher among children of mothers with no formal education”, not that “children of mothers with no formal education are 1.78 times more likely to experience cooccurrence”. Also, the mixing of results from both the bivariate and multivariate analysis in this paragraph is somewhat confusing. Throughout: The way odds ratios are written - “[OR-3.40”, etc. - makes it look like these are negative numbers – I’d suggest removing the hyphens. “socio-economically poorest” vs. “socio-economically poorer” distinction is somewhat confusing – something like “lowest socio-economic quintile” and “second-lowest socio-economic quintile” would be clearer References Reference numbering appears to be off. For example, the first sentence in the introduction is about the global burden of undernutrition, but 2 studies specific to Bangladesh are cited. The Global Nutrition Report 2018 is cited but is 2 years out of date: the 2020 report is now available and can be cited instead/numbers updated.
--	---

REVIEWER	Imam, Abdulazeez
-----------------	------------------

	Medical Research Council Unit The Gambia at the London School of Hygiene and Tropical Medicine, Vaccines and Immunity
REVIEW RETURNED	05-Jun-2021

GENERAL COMMENTS	This article compares differences in risk factors between co-occurrence of undernutrition and single occurrence among Bangladeshi under-fives. This article needs major revision and I have outlined my comments below: Major issues 1.) Authors need to give some background information explaining the biological plausibility for co-occurrence. Why does this occur. Important literature on this topic has been missed and can help develop this paragraph: Briend A, Khara T, Dolan C. Wasting and stunting—similarities and differences: policy and programmatic implications. Food and nutrition bulletin. 2015 Mar;36(1_suppl1):S15-23. 2.) Do authors think they are 're-inventing the wheel'? How do the authors' paper differ from this paper which seems to answer the same question they are attempting to answer using the same BDHS: Chowdhury MR, Khan HT, Mondal MN, Kabir R. Socio-demographic risk factors for severe malnutrition in children aged under five among various birth cohorts in Bangladesh. Journal of Biosocial Science. 2020 Aug 13:1-6. Authors should state if they improve on this paper and how? 3.) Please present a stronger justification for the study. 4.) Methods do not address setting. Ideally, authors should describe Bangladesh in relevant terms - Under 5 population, management for malnutrition is this Community management of acute malnutrition etc...Sample size consideration? 5.) Statistical analysis is not explicit - Please explain how the prevalence of co-occurrence and single dimension were determined. Regression analysis seemed rather cursory. Have the authors heard about backward elimination and forward selection in determining risk factors or predictors in multivariate regression? Please consult a statistician. 6.) What role might multiple testing have played? There are so many tests and p-values 7.) Results table 2 are ambiguous. It is not entirely clear what the authors did here. Are the subgroup prevalences proportions of the total? or are they percentages? what do the P-values test for? Chi-square tests compare proportions, but the individual percentages or prevalences do not add up to a 100% Minor issues 1.) I think the title might need to be changed. The word 'single dimension' does not appear to sit well with undernutrition. 2.) The abstract could be better written. 3.) The introduction does not lead the reader to determine what co-occurrence of malnutrition is in the context of this study. 4.) The opening sentence of the introduction should be referenced appropriately. They do not cite the original articles where these findings are. 5.) some sentences are unclear - Intro line pg 5, 14 to 19, line 54 to 59, pg 6, line 6 to 10, line 29 to 36
--

	6.) Authors should please stick to one term -line 38 Undernutrition or malnutrition 7.) please correct multiple errors - line 59 - duel is dual? cooccurrence is to be spelt as 'co-occurrence'. Discussion page 12, line 22 to 23. Please proof read the manuscript 8.) Please provide a reference for Page 7 line 3 to 88. 99% response rate. 9.) specify the type of bias, pg7 line 27 10.) Occasionally the results prose does not match what is in the table. see example pg8 line 39 - says 7% of mothers, but table reports 3.2%. The same line says 15%, the table reports 14.7%. pg8, line 51 - 51% of children, the table says 52.2%. Please correct this throughout the manuscript. Pg 10 line 8 -higher among children of parent with no formal education when compared to?? 11.) Use of inappropriate words - pg 8 line 48 - "parallel percentage distribution". Discussion line 19 - "a bit worrying" 12.) Table 3 - what determines the use of No as the reference or yes? 13.) Table 3 don't think there is a role for P-values again after reporting AORs 14. Figure 1 and 2 are not legible 15. Authors do not bring out important policy implications for their study. Please see to develop these ideas: Imam A, Hassan-Hanga F, Sallahdeen A, Farouk ZL. A cross-sectional study of prevalence and risk factors for stunting among under-fives attending acute malnutrition treatment programmes in north-western Nigeria: Should these programmes be adapted to also manage stunting?. International Health. 2021 May;13(3):262-71. Wells JC, Briend A, Boyd EM, Berkely JA, Hall A, Isanaka S, Webb P, Khara T, Dolan C. Beyond wasted and stunted—a major shift to fight child undernutrition. The Lancet Child & Adolescent Health. 2019 Nov 1;3(11):831-4. Aryeetey R, Obeng-Amoako GA, Myatt M, Conkle J, Muwaga BK, Okwi AL, Okullo I, Mupere E, Wamani H, Briend A, Karamagi CA. Concurrently wasted and stunted children 6-59 months in Karamoja, Uganda: prevalence and case detection.
--	---

REVIEWER	Walters, Christine Oklahoma State University Oklahoma City
REVIEW RETURNED	07-Jun-2021

GENERAL COMMENTS	Abstract Line 6: delete "current" Line 7: children under 5 years Line 24: bivariate or adjusted odds ratio? Line 28: Instead of saying two out of five, consider using percentage Introduction Line 36: delete "been" Line 43: delete "been" This following sentence needs to be expanded upon: "These indicators of undernutrition are not sufficient to convey the credible estimates of its determinants as many children are suffering from duel or triple conditions of undernutrition." Explain in detail, referencing other researchers and experts in the field and justify the rationale and importance of studying single dimension and occurrence of undernutrition. Methods
--

	Line 26: Sample weights need to be adjusted for in the analysis; please describe in detail how this was done. Discuss how multicollinearity was addressed given the number of variables in your analyses. There are many independent variables; due to the high volume of factors, it would be good to use a theory such as socioecological model or other way or organizing different level of factors. What was the level of significance set at for the bivariate analyses? If the multiple regression analyses were adjusted for confounding variables, AOR (adjusted odds ratio) should be used to represent and discuss results. Discussion Should include specific ideas for future research and detailed recommendations for policy makers. Need to discuss a framework such as UNICEF framework here: https://openi.nlm.nih.gov/detailedresult?img=PMC4428483_fnut-01-00013-g001&req=4
--	---

VERSION 1 – AUTHOR RESPONSE

##Response to Reviewer 1

This is a well-written study comparing the factors associated with a single dimension of undernutrition against the factors associated with the cooccurrence of multiple dimensions of undernutrition among children under 5 in Bangladesh. I would like to especially thank the authors for writing an extremely clear Methods section (with useful references to more details in their supplement).

Major Comments (in order of importance)

Remark: The stated purpose of this study is to compare factors associated with a single dimension of undernutrition against factors associated with cooccurrence. However, this is not exactly what the analytical strategy does – the current analysis actually compares factors associated with single dimension vs. no undernutrition and, separately, factors associated with cooccurrence vs. no undernutrition, not factors associated with single dimension vs. cooccurrence. It is not clear from the analysis whether the differences between single dimension and cooccurrence are statistically significant, and the presentation of the multivariate analysis (Table 3) as adjusted odds ratios also makes it more difficult to compare single vs. cooccurrence, since the table presumably shows odds compared to the reference group of no undernutrition. If the purpose of the study is to compare single vs. occurrence, a different analytical strategy that can incorporate multiple/categorical outcomes, such as multinomial regression, should be considered.

Response: I would like to thank the reviewer for his observation in assessing the comparison. However, the current analysis does not compare factors associated with single dimension vs. no undernutrition and cooccurrence vs. no undernutrition. It compares single dimension vs. others (no undernutrition + co-occurrence) and cooccurrence vs. others (no undernutrition + single dimension). After going through all reviewers' comments on statistical analysis, we have decided to perform stepwise logistic regression analysis.

Remark: Selecting only significant variables from bivariate analysis for multivariate analysis is a commonly-used technique for variable selection, but it can cause potentially important variables to be left out (e.g. variables that are only revealed to be substantive/significant when another variable is controlled for). I urge the authors to consider more robust selection techniques (for example several techniques are discussed here:

<https://scfbm.biomedcentral.com/articles/10.1186/1751-0473-3-17>), either to replace the main multivariate analysis (this would be preferable) or at the very least as a sensitivity analysis to include in the supplement.

o At the very least, if the authors keep the current approach or move it to the appendix, $p < 0.05$ is a very harsh cutoff for variable selection, and it would be better to use a higher threshold.

o The current approach also results in the multivariate analysis for single occurrence using a different set of independent variables than the multivariate analysis for cooccurrence, which seems problematic (e.g. it is difficult to conclude that mother underweight is associated only with cooccurrence and not single dimension when it is not even included in the multivariate analysis for single dimension).

Response: We considered cutoff point 0.25 for the significant p values in bivariate analysis and adjusted more variables in regression analysis.

Remark: Given that other studies have found that distinct factors are associated with stunting vs. wasting vs. underweight, it seems odd to lump the three together as "single dimension" in the analysis. Similarly, might there be different factors associated with being both stunting and underweight vs. being both wasted and underweight vs. all three? I would like to see more Discussion of why this methodological choice was made. If there was not a strong reason for doing this, the authors might want to consider assessing each factor separately (e.g. having 6 categories of undernutrition instead of the 2 currently used).

Response: We have decided not to assess each factor differently. Because, among the six factors, the proportions of four are below 5%. Using different regression analyses in case of below 5% proportion and over 10% proportion will not correctly justify identifying risk factors. Also, it will be challenging to assess and compare risk factors for single dimension and cooccurrence from six different analyses. So, we considered this case as a potential limitation and future scope of research.

Remark: Several variables are missing for over half of the sample (e.g. low birthweight, whether the mother received postnatal or antenatal care). By default, if those variables are included in the logistic regression, missing observations will be dropped entirely, meaning that the logistic regression analysis drops about half of the sample. If the authors used a method for alternative handling of these missing data this should be discussed in the methods. If not, the authors should consider whether/demonstrate that the reduced sample is still nationally representative and does not have substantially different characteristics compared to the full sample. If these conditions do not hold, as a sensitivity analysis the authors should run the analysis on the full sample and remove the independent variables for which there are many missing observations.

o Table 3 should state the sample size to further clarify whether these observations were dropped from the analysis.

Response: Although low birth weight and the mother received postnatal and antenatal care have many missing information, data is nationally representative in most cases. We performed adjusted regression models two times for both cases (single dimension and cooccurrence). We reported the results of regression analysis of all significant variables (from bivariate analysis) except low birth weight; the mother received postnatal and antenatal care in the first model. And in the second model, we entered all significant variables, including low birth weight; the mother received postnatal and antenatal care and reported the results of low birth weight; the mother received postnatal and antenatal care. We provided the note at the bottom of the Table 3.

Remark: I would like to see more Discussion of the importance of assessing factors associated with single dimension vs. cooccurrence of undernutrition. For example, should policy measures target cooccurrence and thus prioritize addressing those risk factors associated with cooccurrence over those associated with a single dimension of undernutrition?

Response: We inserted context-specific recommendations in most of the paragraphs in the discussion section. We added a new paragraph highlighting recommendations/suggestions in the context of the cooccurrence of undernutrition.

Minor Comments

Remark: Abstract:

o This is probably just a typo (as I think all results in the abstract are adjusted odds ratios), but it currently reads as though all of the results are presented as odds ratios except for age group 12-23 months, which appears to be an adjusted odds ratio.

Response: Thanks for finding out the problem. Yes, it is a typo. We corrected.

Remark: Methods:

o First paragraph states that the BDHS "includes data from Bangladeshi adults (both male and female)" – emphasis on adults seems odd given that the focus of this study is on children and data were collected for children too (e.g. anthropometric measurements).

Response: Children are an essential part of the DHS survey, and each survey reveals children's data. So, we mentioned children as vital respondents along with adults.

o Wealth index incorporates access to drinking water and sanitation; both are included in the analysis as distinct, independent variables. It would be better to separate these components so that the independent variables are not overlapping/redundant

For example, we consider only household asset ownership and not access to drinking water/sanitation in the wealth index.

Response: Type of drinking water and sanitation are essential determinants of child malnutrition and wealth index. The categorical structures of access to drinking water and sanitation (yes, no) are different from the type of drinking water and sanitation (improved and unimproved). The standard wealth index's calculation in the DHS was based on access to drinking water, sanitation, and other household assets. It does not affect the models' fit due to their distinct categorical differences—several studies put together these variables.

Remark: Generally, clustering (e.g., standard errors) is distinct from weighting (e.g., using survey weights). Can the authors please clarify this sentence in the statistical analysis subsection of methods? "The data were analyzed by considering cluster effects using survey weights that represent the whole country, both urban and rural areas, to ensure the precision of the estimates"

Response: Cluster means PSU/enumeration areas, strata means urban and rural, and survey weight means sampling weight was considered in this study. We regarded sampling weight in our analysis by controlling cluster and strata. We added and rearranged some information in the data source and statistical analysis section.

Remark: Mother's age at the child's birth might be a more relevant variable to include in the analysis (instead of mother's age when surveyed)

Response: We performed a chi-square test using Mother's age at the child's birth and found it insignificant ($p > 0.05$ for both cases). So we decided not to include this variable to avoid redundancy.

Remark: Results:

o Table 2 should also include the reference group (no undernutrition)

o It would be helpful to add a brief discussion of Figure 2 to the results section.

Response: We added a reference group in Table 2. We also added some discussion regarding Figure 2 in the results section.

Remark: Discussion:

Remark: 2nd sentence "This is a bit worrying figure when comparing it with countries..." Why is this worrying? The burden in Bangladesh is lower than many of those listed (such as India, Yemen, Ethiopia) and this seems like a very random selection of countries to compare against. Requires further explanation.

Response: This section (the first paragraph of the Discussion) has been rearranged. We had to consider these countries randomly as studies regarding this issue were not widely articulated.

Remark: "Four out of ten children of mothers with no formal education experience a cooccurrence of undernutrition." and "Further, children born with low birth weight had a 240% higher risk of having cooccurrence of undernutrition" – where do these numbers come from? They don't appear to be consistent with Table 2. For example, wouldn't 65.5 (37.4% + 28.0%) of children of mothers with no formal education experience cooccurrence?

Response: Some sentences in the second paragraph of the Discussion have been changed in this regard.

Remark: Later - "This study shows that children of mothers with no formal education and underweight mothers were 1.78 and 1.51 times more likely to experience cooccurrence of undernutrition" This sentence conflates relative risks with odds ratios because it comes from the adjusted analysis, which

presents odds ratios. Thus it would be correct to say that "the odds of cooccurrence are 1.78 times higher among children of mothers with no formal education", not that "children of mothers with no formal education are 1.78 times more likely to experience cooccurrence".

Response: These sentences in the second paragraph of the Discussion have been changed based on the recommendation.

Remark: Also, the mixing of results from both the bivariate and multivariate analysis in this paragraph is somewhat confusing.

Response: In the second paragraph of the Discussion, the most influential determinants were addressed based on the bivariate and multivariate analysis. Some sentences have been rearranged to avoid confusion.

Remark: Throughout:

- o The way odds ratios are written - "[OR-3.40", etc. - makes it look like these are negative numbers – I'd suggest removing the hyphens.

- o "socio-economically poorest" vs. "socio-economically poorer" distinction is somewhat confusing – something like "lowest socio-economic quintile" and "second-lowest socio-economic quintile" would be clearer

Response: These suggestions have been addressed in abstract, results, and Discussion.

Remark: References

- o Reference numbering appears to be off. For example, the first sentence in the Introduction is about the global burden of undernutrition, but 2 studies specific to Bangladesh are cited.

- o The Global Nutrition Report 2018 is cited but is 2 years out of date: the 2020 report is now available and can be cited instead/numbers updated.

Response: We added appropriate references in this section.

Response to Reviewer 2

Dr. Abdulazeez Imam, Medical Research Council Unit The Gambia at the London School of Hygiene and Tropical Medicine Comments to the Author:

This article compares differences in risk factors between cooccurrence of undernutrition and single occurrence among Bangladeshi under-fives. This article needs major revision and I have outlined my comments below:

Major issues

Remark: Authors need to give some background information explaining the biological plausibility for cooccurrence. Why does this occur. Important literature on this topic has been missed and can help develop this paragraph:

Briend A, Khara T, Dolan C. Wasting and stunting—similarities and differences: policy and programmatic implications. Food and nutrition bulletin. 2015 Mar;36 (1_suppl1):S15-23.

Response: We added some sentences in this regard in the first paragraph of the Introduction.

Remark: Do authors think they are 're-inventing the wheel'? How do the authors' paper differ from this paper which seems to answer the same question they are attempting to answer using the same BDHS:

Chowdhury MR, Khan HT, Mondal MN, Kabir R. Socio-demographic risk factors for severe malnutrition in children aged under five among various birth cohorts in Bangladesh. Journal of Biosocial Science. 2020 Aug 13:1-6. Authors should state if they improve on this paper and how?

Response: The current study seems to be similar to the above-mentioned paper to some extent. However, the present study highlights the overall undernutrition status based on the most recent data. The reference provided by the reviewer and editor highlighted the severe malnutrition using previous BDHS surveys (excluding BDHS 2017-18) only with a limited number of variables that missed important variables to explain like parental education, mothers nutritional status, birth order, child's birth weight, etc. Also, this article used pooled data (using multiple BDHS surveys) that does not represent the current situation. We added a statement in the last paragraph of the Introduction section in this regard.

Remark: Please present a stronger justification for the study.

Response: We have restructured the introduction section and added more points to justify the study.

Remark: Methods do not address setting. Ideally, authors should describe Bangladesh in relevant terms - Under 5 population, management for malnutrition is this Community management of acute malnutrition etc...Sample size consideration?

Response: We added study setting and management of undernutrition in the method section.

Remark: Statistical analysis is not explicit - Please explain how the prevalence of cooccurrence and single dimension were determined. Regression analysis seemed rather cursory. Have the authors heard about backward elimination and forward selection in determining risk factors or predictors in multivariate regression? Please consult a statistician.

Response: Crosstab analysis and Chi-square test (bivariate analysis) were used to determine the prevalence/percentages and associations (p values) between the selected independent variables and outcomes. Further, all independent variables found significant in bivariate analysis were simultaneously entered into the stepwise (forward selection) multivariate regression models for adjustment to identify risk factors for cooccurrence and the single dimension of undernutrition. We have added this statement in the statistical analysis section and done the analysis accordingly.

Remark: What role might multiple testing have played? There are so many tests and p-values

Response: Variables found significant in the Chi-square test and then adjusted in the Regression model are the popular way of estimating risk factors by assessing the odds ratio.

Remark: Results of Table 2 are ambiguous. It is not entirely clear what the authors did here. Are the subgroup prevalence proportions of the total? Or are they percentage? what do the P-values test for? Chi-square tests compare proportions, but the individual percentages or prevalence do not add up to a 100%

Response: We have included both subgroups for each dependent variable in terms of prevalence analysis using the chi-square test.

Minor issues

Remark: I think the title might need to be changed. The word 'single dimension' does not appear to sit well with undernutrition.

Response: We have changed the title based on the editor and reviewer's comments.

Remark: The abstract could be better written.

Response: We have revised the abstract.

Remark: The Introduction does not lead the reader to determine what cooccurrence of malnutrition is in the context of this study.

Response: We have restructured the Introduction section based on the recommendation.

Remark: The opening sentence of the Introduction should be referenced appropriately. They do not cite the original articles where these findings are.

Response: We have added appropriate references in this section.

Remark: some sentences are unclear - Intro line pg 5, 14 to 19, line 54 to 59, pg 6, line 6 to 10, line 29 to 36

Response: We have restructured the Introduction section and tried to address all these points.

Remark: Authors should please stick to one term -line 38 Undernutrition or malnutrition

Response: We changed malnutrition to undernutrition.

Remark: please correct multiple errors - line 59 - dual is dual? cooccurrence is to be spelt as 'cooccurrence'. Discussion page 12, line 22 to 23.

Response: These corrections have been made.

Remark: Please proofread the manuscript.

Response: English proofreading has been done.

Remark: Please provide a reference for Page 7 line 3 to 88. 99% response rate.

Response: We provided references where necessary and 99% response rate has been addressed.

Remark: specify the type of bias, pg7 line 27

Response: We specified the bias. It was selection bias.

Remark: Occasionally the results prose does not match what is in the table. see example pg8 line 39 - says 7% of mothers, but table reports 3.2%. The same line says 15%, the table reports 14.7%. pg8, line 51 - 51% of children, the table says 52.2%. Please correct this throughout the manuscript. Pg 10 line 8 -higher among children of parent with no formal education when compared to??

Response: We tried to round up the entire numbers with decimal points to their closest number. In the mother's education, it was 7% (both parents uneducated: 3.8% and only mothers were uneducated: 3.2%). We changed the figure 51% male children to 52%. Also, we cross checked all the numbers. We rewrite the sentence in page 9, line 8 as higher among children of uneducated parents (37%) when compare to educated parents (17%).

Remark: Use of inappropriate words - pg 8 line 48 - "parallel percentage distribution". Discussion line 19 - "a bit worrying"

Response: These issues have been addressed.

Remark: Table 3 - what determines the use of No as the reference or yes?

Response: However, it has been chosen randomly, the intensity of risk has been taken into consideration based on the previous research. "No" refers to less risk among the respondents (If it is a reference category), and "yes" refers to high risk and vice versa.

Remark: Table 3, doesn't think there is a role for P-values again after reporting AORs?

Response: We added p values again after reporting AORs in Table 3.

Remark: Figure 1 and 2 are not legible.

Response: After going through all reviewers' comments, we have decided not to remove Figure 1 Figure 2 as it contains some relevant information.

Remark: Authors do not bring out important policy implications for their study. Please see to develop these ideas:

Imam A, Hassan-Hanga F, Sallahdeen A, Farouk ZL. A cross-sectional study of prevalence and risk factors for stunting among under-fives attending acute malnutrition treatment programmes in north-western Nigeria: Should these programmes be adapted to also manage stunting?. *International Health*. 2021 May;3(3):262-71.

Wells JC, Briend A, Boyd EM, Berkely JA, Hall A, Isanaka S, Webb P, Khara T, Dolan C. Beyond wasted and stunted—a major shift to fight child undernutrition. *The Lancet Child & Adolescent Health*. 2019 Nov 1;3(11):831-4.

Aryeetey R, Obeng-Amoako GA, Myatt M, Conkle

J, Muwaga BK, Okwi AL, Okullo I, Mupere E, Wamani H, Briend A, Karamagi CA. Concurrently wasted and stunted children 6-59 months in Karamoja, Uganda: prevalence and case detection.

Response: However, we inserted context-specific recommendations in most of the paragraphs in the Discussion section; we added a new paragraph highlighting recommendations/suggestions in the context of cooccurrence of undernutrition. We inserted these references there.

Response to Reviewer 3

Dr. Christine Walters, Oklahoma State University Oklahoma City Comments to the Author:

Remark: Abstract

Line 6: delete "current"

Line 7: children under 5 years

Line 24: bivariate or adjusted odds ratio?

Line 28: Instead of saying two out of five, consider using percentage

Response: The underlining point in the abstract have been addressed

Response: Introduction

Line 36: delete "been"

Line 43: delete "been"

This following sentence needs to be expanded upon: "These indicators of undernutrition are not sufficient to convey the credible estimates of its determinants as many children are suffering from dual

or triple conditions of undernutrition." Explain in detail, referencing other researchers and experts in the field and justify the rationale and importance of studying single dimension and occurrence of undernutrition.

Response Remark: We have restructured the introduction section based on the recommendation and rewritten the sentence.

Methods

Remark: Line 26: Sample weights need to be adjusted for in the analysis; please describe in detail how this was done.

Response: How sampling weight was calculated from the survey has been described in the method section. Also, stata command to adjust sampling weight has been added.

Remark: Discuss how multicollinearity was addressed given the number of variables in your analyses.

Response: A statement regarding multicollinearity has been added in the Statistical analysis section.

Remark: There are many independent variables; due to the high volume of factors, it would be good to use a theory such as socioecological model or other way or organizing different level of factors.

Response: We have discussed UNICEF framework in the in the "independent variables" section of methodology.

Remark: What was the level of significance set at for the bivariate analyses?

Response: The significance level was set at $p < 0.25$ (2-tailed) in all analyses instead of the traditional cutoff point 0.05, which can cause potentially important variables to be left out. We followed the following reference:

Bursac, Z., Gauss, C.H., Williams, D.K. et al. Purposeful selection of variables in logistic regression. *Source Code Biol Med* 3, 17 (2008)

Remark: If the multiple regression analyses were adjusted for confounding variables, AOR (adjusted odds ratio) should be used to represent and discuss results.

Response: Sorry that we did not understand this remark. We tried to represent and discuss all important variables based on the AOR in the results section.

Discussion

Remark: Should include specific ideas for future research and detailed recommendations for policy makers.

Response: We inserted context-specific recommendations in most of the paragraphs in the Discussion section; we added a new paragraph highlighting recommendations/suggestions in the context of cooccurrence of undernutrition.

Remark: Need to discuss a framework such as UNICEF framework here: https://eur01.safelinks.protection.outlook.com/?url=https%3A%2F%2Fopeni.nlm.nih.gov%2Fdetailresult%3Fimg%3DPMC4428483_fnut-01-00013-g001%26req%3D4&data=04%7C01%7Cmanzur.kader%40ki.se%7C321a21a3b2e746b1be1208

d92a5b6f9b%7Cbff7eef1cf4b4f32be3da1dda043c05d%7C0%7C0%7C637587394629559768%7CUn
 known%7CTWFpbGZsb3d8eyJWIjoiMC4wLjAwMDAiLCJQIjoiV2luMzliLCJBTiI6IjEhaWwiLCJXVCi6
 Mn0%3D%7C1000&sdata=%2B0YUBkYweltspu3W58pm57TGtT3YCDAKULp%2Fh6EK5wU%3
 D&reserved=0

Response: We have discussed the UNICEF framework in the "independent variables" section of methodology.

VERSION 2 – REVIEW

REVIEWER	Ryckman, Theresa Stanford University
REVIEW RETURNED	21-Sep-2021

GENERAL COMMENTS	I would like to thank the authors for their thoughtful responses and revisions to their manuscript. I am satisfied with almost all of the revisions and have just one outstanding comment. Based on the authors' response to my first comment, copied below, I believe that the right 2 columns of Table 4 assess risk factors associated with a single form of undernutrition, compared with both cooccurrence and no undernutrition, which I had misunderstood before. If this is indeed the case, this seems like an odd comparison to make. We would consider a single occurrence to undernutrition to be worse than no undernutrition but better than cooccurrence, and this comparison pools the two together. It is difficult to interpret the results. If the authors would prefer not to switch to an analytical technique that allows for categorical outcomes (such as multinomial regression) I would find it more appropriate to compare single occurrence with just no occurrence and/or with just cooccurrence. I would also recommend that the authors clarify the comparator groups more explicitly in the "Statistical analysis" section of the methods (and possibly in the table notes as well). Here is the response I'm referring to: I would like to thank the reviewer for his observation in assessing the comparison. However, the current analysis does not compare factors associated with single dimension vs. no undernutrition and cooccurrence vs. no undernutrition. It compares single dimension vs. others (no undernutrition + co-occurrence) and cooccurrence vs. others (no undernutrition + single dimension). After going through all reviewers' comments on statistical analysis, we have decided to perform stepwise logistic regression analysis.]
---

REVIEWER	Imam, Abdulazeez Medical Research Council Unit The Gambia at the London School of Hygiene and Tropical Medicine, Vaccines and Immunity
REVIEW RETURNED	18-Sep-2021

GENERAL COMMENTS	I think the authors have gone through great lengths to address the reviewers concerns and should be commended. The manuscript might benefit from further proofreading particularly removing redundant phrases and correcting a few typos. Overall this is a marked improvement to the previous version.
---

VERSION 2 – AUTHOR RESPONSE

Response to Reviewer: 2

Dr. Abdulazeez Imam, Medical Research Council Unit The Gambia at the London School of Hygiene and Tropical Medicine

Remark: I think the authors have gone through great lengths to address the reviewers concerns and should be commended. The manuscript might benefit from further proofreading particularly removing redundant phrases and correcting a few typos. Overall this is a marked improvement to the previous version.

Response: We would like to thank you for this appreciation. We have proofread throughout the manuscript by taking the help of a native English proofreader. Also, we tried to fixed redundant phrases and typos.

Response to Reviewer: 1

Dr. Theresa Ryckman, Stanford University

Remark: I would like to thank the authors for their thoughtful responses and revisions to their manuscript. I am satisfied with almost all of the revisions and have just one outstanding comment. Based on the authors' response to my first comment, copied below, I believe that the right 2 columns of Table 4 assess risk factors associated with a single form of undernutrition, compared with both cooccurrence and no undernutrition, which I had misunderstood before. If this is indeed the case, this seems like an odd comparison to make. We would consider a single occurrence to undernutrition to be worse than no undernutrition but better than cooccurrence, and this comparison pools the two together. It is difficult to interpret the results. If the authors would prefer not to switch to an analytical technique that allows for categorical outcomes (such as multinomial regression) I would find it more appropriate to compare single occurrence with just no occurrence and/or with just cooccurrence. I would also recommend that the authors clarify the comparator groups more explicitly in the "Statistical analysis" section of the methods (and possibly in the table notes as well).

Here is the response I'm referring to:

"I would like to thank the reviewer for his observation in assessing the comparison.

However, the current analysis does not compare factors associated with single dimension vs. no undernutrition and cooccurrence vs. no undernutrition. It compares single dimension vs. others (no undernutrition + co-occurrence) and cooccurrence vs. others (no undernutrition + single dimension). After going through all reviewers' comments on statistical analysis, we have decided to perform stepwise logistic regression analysis.]"

Response: We highly appreciate this comment. As per the recommendation, we have tried to clarify this aspect by re-categorizing the dependent variable as no undernutrition vs. cooccurrence and no

undernutrition vs. single dimension, particularly for adjusting regression analysis. Further, we have repaired the results, discussion, and abstract section based on the analysis.